# Effect of Plasma Activated Water on Selected Chemical Compounds of Rocket-Salad (*Eruca sativa* Mill.) Leaves

**DOI:** 10.3390/molecules26247691

**Published:** 2021-12-20

**Authors:** Doaa Abouelenein, Simone Angeloni, Giovanni Caprioli, Jessica Genovese, Ahmed M. Mustafa, Franks Kamgang Nzekoue, Riccardo Petrelli, Pietro Rocculi, Gianni Sagratini, Silvia Tappi, Elisabetta Torregiani, Sauro Vittori

**Affiliations:** 1School of Pharmacy, University of Camerino, Chemistry Interdisciplinary Project (CHIP) via Madonna delle Carceri, 62032 Camerino, Italy; doaa.abouelenein@unicam.it (D.A.); simone.angeloni@unicam.it (S.A.); giovanni.caprioli@unicam.it (G.C.); ahmed.mustafa@unicam.it (A.M.M.); astride.kamgang@unicam.it (F.K.N.); gianni.sagratini@unicam.it (G.S.); elisabetta.torregiani@unicam.it (E.T.); sauro.vittori@unicam.it (S.V.); 2Department of Pharmacognosy, Faculty of Pharmacy, Zagazig University, Zagazig 44519, Egypt; 3Department of Agricultural and Food Sciences, University of Bologna, Piazza Goidanich, 60, 47522 Cesena, Italy; jessica.genovese3@unibo.it (J.G.); pietro.rocculi3@unibo.it (P.R.); silvia.tappi2@unibo.it (S.T.); 4Interdepartmental Centre for Agri-Food Industrial Research, University of Bologna, Via Q. Bucci 336, 47522 Cesena, Italy

**Keywords:** plasma activated water, arugula, rocket volatile profile, phytosterols, β carotene, luteolin, chlorophyll a, chlorophyll b

## Abstract

Plasma activated water (PAW) has proven to be a promising alternative for the decontamination of rocket leaves. The impact of PAW on the volatile profile, phytosterols, and pigment content of rocket leaves was studied. Leaves were treated by PAW at different times (2, 5, 10, and 20 min). Compounds of the headspace were detected and quantified using GC–MS analysis. A total of 52 volatile organic compounds of different chemical classes were identified. Glucosinolate hydrolysis products are the major chemical class. PAW application induced some chemical modifications in the volatile compounds. Changes in the content of the major compounds varied with the increase or decrease in the treatment time. However, PAW-10 and -2 were grouped closely to the control. A significant decrease in the content of β-sitosterol and campesterol was observed after PAW treatment, except for PAW-10, which showed a non-significant reduction in both compounds. A significant increase in β carotene, luteolin, and chlorophyll b was observed after the shortest treatment time of PAW-2. A reduction in chlorophyll content was also observed, which is significant only at longer treatment, or PAW-20. Overall, PAW has proven to be a safe alternative for rocket decontamination.

## 1. Introduction

*Eruca sativa* Mill, also known as arugula salad, cultivated rocket, rucola, or roquette, is gaining popularity as a fresh cut ready-to-eat product. Plant leaves are commonly sold in whole bags, mixed salad bags, or as gourmet micro-leaves [1]. Rocket is considered as a medicinal plant. In fact, it was known for its aphrodesiac effect, which was reported in ancient texts. It has been also reported to have potential antioxidant, immune boosting, and anti-inflammatory effects [2]. Moreover, Azarenko et al. [3] focused on the application of erucin (a major compound in rocket) to human breast adenocarcinoma cells. Also, rocket leaves were reported to increase plasma nitrate and nitrite which could significantly reduce blood pressure [4]. Rocket is valued for its sensory and nutritional properties given by the volatile organic compounds (VOCs) found in the plant. VOCs comprise glucosinolate hydrolysis products (GHPs), alcohols, ketones, aldehydes, fatty acids, esters, and alkanes [1,5]. It is widely accepted that the rocket’s distinctive aroma and flavour are produced by GHPs. This aroma could influence the sensory attributes perceived by the consumers and determine whether the product will be accepted or rejected, influencing the product’s re-purchase. Rocket-salad leaves are also rich in phytonutrients such as polyphenols, glucosinolates, phytosterols, vitamins, carotenoids, and fibres [6,7,8,9], which are responsible for their health promoting effects.

Minimally processed vegetables may suffer the outbreak of foodborne diseases and the deterioration of their chemical and physiological properties. As a result, it is critical to develop effective methods for prolonging the fresh state and maintaining the content and activity of bioactive compounds in these products. This would improve their marketability and have a positive impact on human health [10,11]. Traditional processing methods have led to good inactivation of microorganisms, but can alter the sensory and nutritional properties of foods due to heat exposure or the addition of preservatives. Therefore, new emerging technologies have been introduced to improve productivity by increasing the shelf life of foods without changing their nutritional and organoleptic properties [12].

Cold plasma is a new, environmentally friendly, and chemical-free non-thermal disinfection technology. Recent studies on the plasma antimicrobial activity on food showed satisfactory results. On the other hand, a few researchers reported some negative effects, such as bioactive compounds degradation and colour loss after surface treatment. To overcome these problems, plasma activated water (PAW), in which an acidic environment is created resulting in the formation of reactive oxygen and nitrogen species (ROS & RNS), has been used. Therefore, PAW has a different chemical composition than water and can be regarded as an alternative microbial disinfection method [13].

PAW has proven to be a promising strategy for the decontamination of rocket-salad leaves and is regarded as a promising alternative to hypochlorite treatment, with the advantage of having a less negative impact on the environment and the health of consumers. Shorter PAW treatments have shown a significant reduction in populations of Enterobacteriaceae and psychrotrophic bacteria, and higher inactivations were obtained for all studied microbial groups after 2 min of treatment [14]. However, little information on the effect of PAW on the content of the bioactive compounds of rocket salad has been reported. For this reason, the present work aims to evaluate, for the first time, the effect of different processing times (2, 5, 10, and 20 min) of the PAW technique on the rocket-salad’s bioactive compounds (volatile profile, phytosterols, carotenoid, and chlorophyll contents) in order to evaluate the use of this technology as an alternative decontamination method.

## 2. Results and Discussion

### 2.1. Effect of PAW Treatment on the Main VOCs in Rocket Leaves

Detection of the main volatile compounds in rocket samples was performed, which is illustrated by the chromatogram presented in Figure 1. Table 1 describes the VOCs identified in the control and PAW-treated rocket samples and their relative abundances, together with their experimental retention indices (RI). The volatile composition of rocket samples was in a good agreement with previously published data [1,5,15,16]. In total, 52 compounds were identified, and the most predominant class was glucosinolate hydrolysis products (GHPs) (7 compounds) followed by other sulphur containing metabolites (6 compounds). In addition, 13 ketones, 13 aldehydes, 4 fatty acids and esters, 8 alcohols, and 1 compound of the class of alkanes were identified.

#### 2.1.1. Glucosinolate Hydrolysis Products (GHPs)

Glucosinolates (GSLs) in rocket leaves are not only the major class in terms of their concentration, but also the main contributors to the bitter and pungent taste, which is quickly formed when the rocket leaf tissues are crushed. GSLs are normally hydrolysed by the brassica enzyme myrosinase into GHPs; e.g., isothiocyanates (ITCs) or nitriles which are generally considered to be responsible for the pungency and Brassicaceae-like aroma [15,17]. However, the GHPs, whether ITCs or nitriles, retain the R group of GSLs, which has an impact on their bioactivity [18]. Therefore, monitoring of GHPs could be of use as a marker for nutritional composition and sensorial quality of rocket.

As shown in Table 1, seven kinds of hydrolysates were detected in the control rocket sample, including: methyl thiocyanate, two ITCs (1-butene 4-isothiocyanate, 4-methylthio-butyl isothiocyanate), and four nitriles (4-methylthio butanenitrile, 5-methyl hexanenitrile, heptanenitrile, 5-methylthiopentanonitril). All these compounds have been reported before in rocket leaves [5,15,17,19]. As shown in Figure 1, erucin nitrile (5-methylthiopentanenitrile) and erucin (4-methylthiobutyl isothiocyanate) were the major compounds detected in the chromatogram of the control rocket sample. As reported by [20,21], erucin and erucin nitrile are the degradation products of glucoerucin (one of the major GSLs found in rocket leaves) (Figure 2). Besides being the major VOC in rocket leaves [15,19], erucin has been reported as one of the most potent odour-active compounds in rocket, being associated with radish and typical rocket aroma [15,17]. However, erucin nitrile was reported as a major compound in samples obtained from dried plant material [5]. This suggests that drying of the plant material, in our case, contributed to the degradation of glucoerucin towards erucin nitrile being the major VOC in the control sample, followed by the erucin (Table 1).

Non-significant changes in the total relative abundance of this class in all PAW treated samples were observed. Additionally, the process carried out at 2 and 10 min did not show any significant changes in the individual contents of this class. On the other hand, significant qualitative and quantitative changes were observed for both PAW-5 and -20 samples. Regarding PAW-5, three compounds could not be identified, which are 5-methyl hexanenitrile, heptanonitril, and 4-methylthio butanenitrile. A significant decrease in the former two compounds was also observed in PAW-20. Moreover, a significant decrease in the major VOC erucin nitrile was also observed in both PAW-5 and PAW-20 samples, which was accompanied by a significant increase in the relative percentage of erucin when compared to the control, as shown in Figure 3. Interestingly, it can be concluded that the abandonment of nitrile production in favor of ITCs occurred in these two samples. It was previously reported that different factors could affect the yield and abundance of GHPs from rocket leaves. These may include pH, the solvent used for extraction, the method for leaf homogenization, liquid or headspace extraction, and sample state (e.g., fresh or dried). As GHPs’ profiles change rapidly, it is difficult to compare absolute quantities between various studies [2]. One study reported that the amount of erucin and erucin nitrile produced during this hydrolysis is pH dependant [22]. Bell et al. [23] also reported a low amount of nitrile compounds detected in rocket leaves after processing, which was suggested to be a cause of the acidity of hydrolysis conditions. Another study on the hydrolysis of the GSLs to their nitriles using beneficial bacteria confirmed that aerobic and anaerobic conditions favored the production of erucin nitrile, but for Enterobacteriaceae in aerobic conditions, only trace amounts of erucin nitrile were produced [18]. This may explain the reciprocal transformation among erucin and erucin nitrile during glucoerucin hydrolysis in PAW treated samples. However, this effect did not seem to be time-dependent.

Despite the low content in terms of the overall volatile profile, the compound 1-butene 4-isothiocyanate was reported to have descriptions of typical pungency in rocket at high intensities [17]. The compound also showed a significant increase in both PAW-5 and -20 samples. Generally, along with their contribution to the rocket aroma, ITCs are suggested to have cytotoxic activity against the most common cancer types [21]. Our data infer that the retention of ITCs in PAW-2 and PAW-10 samples, or their significant increase in PAW-5 and PAW-20, will have important implications for health benefits to the consumer. Cold plasma treatment has also previously been reported to increase total ITC content in green mustard seeds [24]. Since ITCs can survive during PAW processing, this may suggest that PAW processing can enhance this property of rocket leaves.

#### 2.1.2. Sulphur Compounds

Together with the ITCs, other sulphur containing compounds have been detected in rocket leaves, including a high content dihydro-2*H*-thiopyran-3(4*H*)-one, which has been detected before in rocket salad by [17]. Generally, positive correlations have been reported between ITCs and sulphur compounds detected in rocket with bitter, peppery, mustard, and initial heat mouthfeel characters [16]. A significant reduction in the relative abundance of the sulfur-containing compound is also evident in the reduction in dihydro-2*H*-thiopyran-3(4*H*)-one in all samples after treatment without significant differences among treatment times. Bußler [25] has reported that sulphur-containing compounds, including sulphur containing aromatic amino acids, are preferred for attacks of ROS released from the plasma treatment, hence it is possible to hypothesise oxidation of these components due to the reactive species present in PAW.

#### 2.1.3. Ketones

Ketones are reported to have an important role in plant defense. They were previously reported to contribute to the sensory characters of rocket, as they are correlated with the pleasant odours [16]. Among the identified ketones, 3,5-Octadien-2-one and (3*E*,5*E*)-3,5-Octandiene-2-one were described to impart a pungent green aroma of medium intensity in Brassicaceae species [17]. In addition, 6-Methyl-5-hepten-2-one was previously identified in rocket leaves by [15,17], and hexahydrofarnesyl acetone (phytone), a very common ketone in Brassicaceae plants that resulted from the oxidative degradation of the diterpene alcohol (*E*)-phytol that occurs as a side chain of chlorophyll a [26], were detected. Among the identified ketones, another group, called volatile norisoprenoids was also identified in rocket samples, such as (*E*)-β-ionone, β-ionone-5,6-epoxide, (*E*)-geranylacetone, and dihydroactinidiolide, that were detected in all rocket samples. Bell, Kitsopanou, Oloyede, and Lignou [17] have reported the presence of geranylacetone (0.1 %) in rocket leaves. The major ketone in all rocket samples was β-ionone, also previously detected in rocket leaves by [5,15]. Moreover, dihydroactinidiolide was detected in Brassicacea by Oulad El Majdoub et al. [27].

Except for PAW-5, which showed a significant increase in the total relative abundance of ketones, other PAW treatments showed non-significant changes compared to the untreated sample. Our results are in good agreement with some previous results, such as those of Korachi et al. [28], who reported non-significant changes in the total composition of ketones in milk following cold plasma treatment. Regarding the individual compounds, PAW-20 and PAW-10 did not induce any significant changes except for the β-ionone-5,6-epoxide, which increased significantly in PAW-10 samples. Moreover, a significant increase was observed in the relative abundance of 6-methyl-5-hepten-2-one and β-ionone-5,6-epoxide in PAW-2 samples, however, 2,5-dimethyl-3-hexanone could not be identified. PAW-5 showed a significant increase in 2,5-dimethyl-3-hexanone, 3-octen-2-one, 3,5-octadien-2-one, (*E*)-β-ionone, and β-ionone-5,6-epoxide, which consequently led to the significant increase in the total ketone content of this sample. Furthermore, Liu et al. [29] detected higher contents of geranylacetone in brown rice upon processing with cold plasma. A reason for the increase in total content of ketones in PAW-5 could be the lipid oxidation that may have occurred in this sample, which can lead to further formation of the secondary lipid oxidation products epoxides, aldehydes, dimers, or ketones [30].

#### 2.1.4. Aldehydes

The aldehydes which were detected in the rocket samples were 2-methyl propanal, 2-methyl butanal, 3-methyl butanal, pentanal, hexanal, 2-hexenal (*E*), octanal, nonanal, 3-furfural, benzaldehyde, β-cyclocitral, benzeneacetaldehyde, and 2-methyl benzaldehyde. It is worth mentioning that aldehydes showed a high degree of association with taste, flavour, and mouth-feel traits in rocket [16]. (*E*)-2-hexenal and hexanal were associated with the green aroma impression of the rocket-salad. Also, herbal aromas are caused by (*E*)-2-hexenal. Floral–fruity odour notes can be associated with nonanal. At the same time, nutty and almond-like odour impressions are known from furfural and benzaldehyde. Hexanal and nonanal compose the fatty side-notes [19]. Overall, non-significant changes were observed in the total content of aldehydes in PAW-2, -10, and -20. However, a significant increase (*p* < 0.05) in the level of total aldehydes was observed only in PAW-5 samples. A significant increase was observed in the content of 2-methyl propanal, 2-methyl butanal, pentanal, hexanal, octanal, 3-furfural, benzaldehyde, and β-cyclocitral (*p* ≤ 0.05). In addition, a non-significant increase was revealed for all the other detected aldehydes (*p* ≥ 0.05). The results evidenced that plasma treated cells in PAW-5 accumulated higher amounts of several aldehydes compared to the controls. Previous studies have also reported an increase in aldehyde content in guava-flavored whey beverages [31] and milk [28] after plasma treatment. This increase in these aldehydes could be attributed to the degradation of several unsaturated fatty acids found in rocket [1] by auto-oxidation and/or the spontaneous decomposition of hydroperoxides. Such degradation could be the result of the damaging effect of reactive species produced by the plasma, which can initiate lipid peroxidation and produce hydroperoxide, and can then be converted to secondary oxidation products such as aldehydes or shorter fatty acyl compounds. [32,33]. However, further studies are needed to confirm these assumptions.

#### 2.1.5. Alcohols

Alcohols may be formed by the decomposition of fatty acid hydroperoxides or the reduction in aldehydes [34]. They are used as a defensive mechanism of plants and are often responsible for the ‘cut grass’ aroma found in leafy vegetables [1,35]. In this study, a significant increase was observed in total relative abundance of alcohol with PAW-2 and PAW-10 samples. The content of hex-3-en-1-ol, which was detected as a major alcohol in Eruca spp [5,16,19], increased after PAW application. Additionally, PAW treatment increased the content of hexan-1-ol. These two compounds are typical green leaf volatiles (GLVs) produced naturally in plants. They are derived from linoleic acid through the lipoxygenase (LOX) enzymatic route [36,37]. The pathway produces hexanal and hex-3-enal by the oxygenation of linoleic acid through the catalysis of LOX, which, by further reduction in the aldehydes, produces hexan-1-ol and hex-3-en-1-ol. According to our results, an increase in both compounds was observed with all PAW treatments when compared to controls, with a non-significant decrease in hex-3-en-1-ol in PAW-5 and -20. These results agreed with the previous study that reported the increase in the contents of hexan-1-ol and hex-3-en-1-ol observed in camu-camu pulp. The results were explained that either the LOX enzyme was activated by plasma application or the oxidation of linoleic acid was catalysed by the reactive oxygen species formed during plasma generation [38]. Further, 1-penten-3-ol, which is significantly correlated with sweet attributes in rocket [16], was increased in all PAW-treated samples, with a significant increase in PAW-2. Moreover, the significant increase in the content of pentan-1-ol and phenylethyl alcohol in PAW-2 samples was also observed. Octan-1-ol was significantly increased in PAW-20 samples. On the other hand, phenylethyl alcohol was not detected in PAW-5 samples.

Interestingly, profiles of VOCs in all four PAW-treated rocket samples showed some differences, such that hierarchical clustering analysis (HCA) was performed on the data of the 52 compounds detected in all samples: control, PAW-2, PAW-5, PAW-10, and PAW-20, in order to have a conclusive idea as to the effect of PAW treatment on the volatile profile. As shown in the dendrogram (Figure 4), the five samples were sorted into three logical classes. PAW-10 was first grouped with the control sample with a 99.7% similarity level. The PAW-2 sample was then isolated but still grouped closely with a high similarity level (99.1%), which confirms that PAW did not induce significant changes in the volatile profile of these two samples. On the other hand, the profiles of PAW-5 and PAW-20 samples were grouped together, but were distinct from that of the control (similarity level 82.7%). However, the results indicated a non-significant effect of the PAW-treatment on rocket-salad’s volatile profile. Further research is needed to better understand the effect of PAW-processing time on the volatile profile of rocket samples.

### 2.2. Effect of PAW Treatment on the Main Phytosterols in Rocket Leaves

Phytosterols are a class of lipids which refer to steroidal compounds, physiologically and structurally similar to cholesterol. They are naturally present in foods of plant origin and exhibit blood LDL cholesterol-lowering properties. Moreover, they exert anti-cancer, hepato-protective, and anti-inflammatory properties [39]. Recently, the oxidation of food lipids due to plasma reactive species has received much attention, since literature reveals that cold plasma could induce lipid oxidation in different types of foods, including rice [40], wheat flour [41], and olive oil [42], via the action of reactive species. The problem of lipid oxidation severely affects the quality of food products, and sometimes limits their shelf-life. It also causes loss of flavour or development of off-flavours, loss of colour andnutrient value, and the accumulation of compounds which may be detrimental to health [43]. For this reason, it is important to study the impact of a novel technology such as PAW on the content of these micronutrients.

β-sitosterol and campesterol are the main phytosterols reported in rocket leaves [6]. The content of each compound was determined by HPLC-DAD in control and PAW-treated samples, and the results are summarized graphically in Figure 5. The results demonstrate that PAW treatment induced a significant reduction in both β-sitosterol and campesterol contents in all treatments, except for PAW-10, which showed a non-significant decrease. It is noteworthy that no studies were performed on the effect of PAW treatment on phytosterols in rocket, though relatively little information exists on the effect of plasma treatment on phytosterols’ content in other plants. For example, Yodpitak et al. [44] reported a significant decrease in total phytosterol content in one of the brown rice cultivars, while the other cultivars showed non-significant changes after cold plasma treatment. This significant reduction in both β-sitosterol and campesterol contents could be due to an autoxidation process promoted by the plasma reactive species, as also confirmed by the increase in the aldehydes, alcohols, and ketones as the main volatile compounds generated in the lipid oxidation process [43]. However, also in this case, the changes are not strictly proportional to treatment time.

The composition of PAW has been partly determined by [14] through the measurement of H_2_O_2_, NO_2_^−^, and dissolved O_3_, not only after the discharge, but also during the following 20 min. As expected, the concentration of the considered species decreased over time due to their high reactivity, particularly for short-lived compounds such as peroxides, while nitrites decreased only slighlty. However, these are only a part of all the chemical reactive components present in PAW, whose chemsitry is very complex and changing over time. Moreover, in many of the published works on the effect of PAW on food products, information about plasma chemistry is often lacking or incomplete. Therefore, it is difficult to identify which are the species directly responsible for the observed results.

### 2.3. β-Carotene and Lutein Contents

Carotenoids are lipophilic compounds with several conjugated double bonds and 40 carbon molecules. They are classified chemically as xanthophylls, which have one or more oxygen groups (e.g., lutein and zeaxanthin), and carotenes, which are non-oxygenated (e.g., lycopene and β-carotene). Long-chain carotenoids are significantly more susceptible to oxidation and isomerization, which can happen during processing and storage [45]. Lutein and β-carotene have been reported to be the most abundant carotenoids in rocket leaves [46,47,48,49]. A higher content of β-carotene than that of lutein in both garden and wild rocket samples has been previously reported [46], however, lutein was reported at higher concentrations in rocket leaves [47,48]. The content of lutein in control rocket samples was 37.40 mg/100 g D.W., as shown in Figure 6. The β-carotene content was 36.31 ± 3.41 mg/100 g D.W., which is consistent with the findings of [49], who reported a β-carotene content of 36.00 ± 0.01 mg/100 g in dried rocket leaves. However, both β-carotene and lutein contents found in this study were higher than the values reported by previous studies [46,47,48].

Figure 6 shows the effect of PAW treatment on the carotenoid content of control and treated samples. Similar behaviour was observed for the content of both carotenoids (β-carotene and lutein) after short PAW treatment, in which an increase in the contents of both carotenoids was observed after PAW treatment with PAW-2, -5, and -10; this increase was significant for both carotenoids after the shortest treatment time (PAW-2). Subjecting the samples to a longer processing time (PAW-20) lead to a non-significant reduction (*p* ≥ 0.05) in lutein content. Generally, carotenoids are stored in chromoplasts, which work together with the cell walls and cell membranes, and act as natural barriers to their release. Disrupting or weakening of these natural barriers was previously reported as crucial for increasing the bio-accessibility of carotenoids. As a result, food processing has been regarded as an important tool for this purpose [45]. The exact biochemical pathway that leads to a rise in the carotenoid content is still unknown, but reactive plasma species have previously been reported to react by breaking the bond between carotenoid molecules and cell membranes, resulting in an increase in the concentration of free carotenoids [50]. Exposure to electrically charged species from cold plasma has also been reported to lead to a certain degree of electroporation [50], that, in turn, can promote changes to the hydrophobic and hydrophilic properties of the membrane and contribute to the release of fat-soluble compounds bound to the cell pulp membrane (Martínez, et al. [51]). The longer processing times tended to cause a non-significant reduction in lutein content due to the higher concentration of reactive species that may have accumulated in the samples [50]. Thus, it could be assumed that the carotenoids’ radical scavenging behaviour may contribute to their breakdown in the presence of a higher content of free radicals and ions [52]. Contrary to our results, a reduction in the content of carotenoids was previously reported in kiwi [53], pumpkin puree [54], guava beverages [31], and tomato [55] post plasma treatment. However, the plasma generation systems, sources, and modes of application were different compared to those used in the present research.

### 2.4. Chlorophylls (Chl) Content

Table 2 shows the values of chlorophyll a, b, and total chlorophyll measured in the rocket samples treated with PAW and compared to the untreated sample. The total chlorophyll content of the untreated sample was 243 mg/100 D.W. This value is lower compared to the one reported by [46] for both garden and wild rocket. However, the chlorophyll content can vary quite substantially depending on the cultivar and on the physiological state of the tissue.

After PAW immersion, a slight decrease was observed in Chl-a content, but it was only significant after the longer treatment (PAW-20), where a 30% reduction was observed. Instead, Chl-b showed a slight increase after the shortest treatment (PAW-2) (+23%) but then remained relatively constant during the longer immersion times. The total chlorophyll content showed significantly lower values compared to the untreated sample after 5 and 20 min.

On one side, the degradation of chlorophyll can be favoured by the presence of oxygen radicals that promote the oxidation of the pigments. On the other side, the increase in Chl-b after 2 min of immersion could be explained by an increased extractability due to cell structure breakdown. Hence, the total amount of chlorophylls can be considered a result of the balance between higher extractability and oxidative degradation. A reduction in chlorophyll content was also observed by [53] on kiwi fruit slices exposed to a DBD plasma treatment. The authors observed a 15% reduction on Chl-a, which was attributed to the Type II breakdown mechanism (chemical breakdown). This variation, however, did not affect the colour of the samples. However, a variation in the colour parameters in rocket samples immersed in PAW has been previously reported, in particular a decrease in the luminosity of red and green indexes was observed for all samples, from 2 to 20 min immersion [14]. We can therefore assume that this variation could be partly related to the partial degradation of chlorophyll, which is the main pigment in rocket leaves.

As shown previously [14], increasing treatment time from 5 to 20 min did not bring any significant improvement to the microbial decontamination of rocket leaves for the considered microbial species. Therefore, on one side, it could be considered useless to prolong the treatment above 5 min; however, as the results obtained in the present research have shown, the choice of the treatment time should also take into account the effect on the nutritional parameters in order to maximise the overall quality of the final product.

## 3. Materials and Methods

### 3.1. Raw Materials

Fresh rocket leaves were acquired in the local market (Cesena, Italy) and transported to the laboratory, where they were kept at 2 ± 1 °C in a refrigerated cell for up to 24 h until processing. Intact leaves free of defects were selected and divided into sub-samples to be subjected to the different treatments.

### 3.2. PAW Generation

PAW was obtained through the prototype described by [14], exposing distilled water for 4 min to plasma through a novel plasma source (Figure 7) composed by a stainless-steel pin-electrode connected to a microsecond pulsed generator (AlmaPulse, AlmaPlasma s.r.l.). Peak voltage and frequency used for plasma generation were 9 kV and 5 kHz, respectively, and the pulse’s duration was 200 µs. A volume of 450 mL of water, contained in a borosilicate Erlenmeyer flask on a stirrer (IKA Magnetic Stirrers RCT basic), was directly connected to the ground and exposed to plasma. A magnetic stirrer was set at the speed of 200 rpm. Cold plasma was generated in the air gap (5 mm) between the tip of the high voltage electrode and the water surface

The calculated average discharge dissipated power was 480.95 ± 33.42 W. The pH of PAW after the discharge was 3.3. Previously determined concentrations of H_2_O_2_, NO_2_^−^, and dissolved O_3_ were respectively 4.5 ± 0.1 mg/L, 30.4 ± 0.9 mg/L, and 0.3 ± 0.1 mg/L.

### 3.3. PAW Treatment

Immediately after the water treatment, the rocket samples were immersed in PAW for 2, 5, 10, and 20 min with a product:liquid ratio of 1:20 at room temperature. During immersion, samples were continuously agitated in an orbital agitator. After dipping, rocket leaves were removed from PAW and blotted with absorbent paper for the removal of the excess liquid. For each treatment time, two independent treatments were carried out. Untreated rocket leaves were considered as control. After each treatment time, rocket samples from both washing replicates were immediately freeze-dried; then, the obtained samples were stored at −20 °C until analysis.

### 3.4. Volatile Organic Compounds (VOCs) Analysis

#### 3.4.1. Headspace Solid-Phase Microextraction (Hs-Spme)

First, 400 mg of lyophilized rocket samples were weighted into a 20 mL vial, sealed with a screw cap with a PTFE septum, and equilibrated at 80 °C for 30 min. The incubation of the sample was completed under agitation (250 rpm, 5 s of on-time and 2 s of off-time) and extracted using PDMS/DVB 65 μm SPME fibre coatings. The fibre was conditioned for 15 min at 250 °C and then exposed to the sample headspace at a penetration depth of 45 mm with a speed of 20 mm/s. The temperature was kept at 80 °C throughout the extraction of the volatile compounds for 15 min without agitation. After the extraction, the volatiles were directly desorbed on the GC liner and maintained at 250 °C for 2 min for fibre reconditioning.

#### 3.4.2. Gc–Ms Analysis

Samples were analysed using an 8890-gas chromatograph (GC) from Agilent, equipped with a PAL RTC 120 autosampler and a 5977B mass spectrometer (MSD) (Agilent, Santa Clara, CA, USA). The ionization was obtained by using an electron ionization source (EI). The injector temperature was set at 250 °C, and the liner used was recommended for SPME injection, namely, the Inlet liner, Ultra Inert, splitless, straight, 0.75 mm id, from Agilent. The gas carrier was helium at a flow rate of 1.0 mL/min. The separation of target molecules was established onto on a DB-Wax column (60 m, 250 μm i.d., 0.25 μm film thickness). The oven temperature program started at 35 °C for 3 min, before increasing to 180 °C at 3 °C/min and from 180 to 210 °C at 15 °C/min, and the final temperature (250 °C) was held for 10 min. The acquisition was carried out in SCAN mode (35–450 *m*/*z*). The compounds’ identification was performed by comparison of their mass spectra and their experimental retention indices (RI) with data of the NIST library (US National Institute of Standards and Technology) and with those available in the literature [1,5,15,16]. The relative percentages of the individual components were calculated based on GC peak area, which was obtained by dividing the area of each component by the total area of all separated components. Percentage values were the means of two replicates for each sample. Data results were managed using MSD ChemStation Software (Agilent, Version G1701DA D.01.00, Santa Clara, CA, USA).

### 3.5. Phytosterol Analysis by HPLC–DAD

#### 3.5.1. Extraction of Phytosterols from Rocket Samples

The extraction was completed as proposed previously by Nzekoue et al. [56], with slight modifications, where 100 mg of each sample was mixed with 1N HCl (1 mL) and water (3 mL), then sonicated for 10 min (59 Hz). After sonication, the samples were saponified for 40 min and extracted after cooling with hexane (10 mL × 3), then collected and dried with a rotary evaporator. Subsequently, the dry extracts were dissolved in 1 mL of hexane. Dansylating was used to derivatize the extracted phytosterols in which 1 mL of the hexane extract was mixed with 20 µL of hexaconazole (500 µg/mL), then dried under nitrogen and redissolved with 2 mL of dichloromethane containing dansyl chloride and DMAP, both at a concentration of 8 mg/mL. Then, the sample was dried under nitrogen and dissolved in 1 mL of acetonitrile. Samples were sonicated and filtered for HPLC analysis.

#### 3.5.2. HPLC–DAD Analyses

The dansylated phytosterols were detected using a 1260 Infinity liquid chromatography system (Agilent Technologies, Santa Clara, CA, USA) with an autosampler, quaternary pump, and a diode array detector (DAD). The sample injection volume was 20 μL and the separation of analytes was performed on a Gemini C18 analytical column (250 × 3.0 mm, 5 μm) preceded by a security guard column C18 (4 × 3 mm, 5 μm) (Phenomenex, Torrance, CA, USA). Methanol (100%) was used as a mobile phase at a flow rate of 0.5 mL/min. The elution was performed in isocratic mode, and phytosterols were detected and quantified at λ 254 nm [56].

### 3.6. Determination of β-Carotene and Lutein Contents by HPLC–DAD

Both carotenoids were extracted according to the method described by [47], with slight modifications, where 50 mg of lyophilized rocket sample were rehydrated with 5 mL ethanol containing 1 mg/mL of butylated hydroxytoluene (BHT). Then, 1 mL of a 50% (*w*/*v*) methanolic KOH solution was added and rocket extracts were saponified for 10 min at 85 °C (in the dark). Samples were cooled in an ice bath and 2 mL of ice-cold water was added. The suspensions had been extracted two times with 2 mL of hexane by vigorous shaking, then centrifuged at 5000 rpm for 10 min at room temperature. The upper hexane layers were separated and evaporated to dryness. Dried extracts were redissolved in 1 mL of an acetonitrile–methanol–dichloromethane solution (60:30:10 *v*/*v*) and filtered before injection. Carotenoid concentrations were determined by HPLC Agilent 1260 infinity II series (Santa Clara, CA, USA) using the method of [57], with some modifications. The analyses were carried out on a Chromolith RP-18e analytical column (100 × 3 mm I.D., macropore size 2 µm, mesopore size 13 nm) from Merck (Darmstadt, Germany). A gradient was used, with a mobile phase composed of MilliQ water (A), acetonitrile (B), and 2-propanol (C); the solvent composition was changed as follows: 0–2 min, 30:70 A/B (*v*/*v*); 2–5 min, 100% B (*v*/*v*); 5–11 min, 80:20 B/C (*v*/*v*); 11–10 min, 100% B (*v*/*v*); 12–14 min, 30:70 A/B (*v*/*v*) at a flow rate of 0.8 mL/min. The detection of carotenoids was carried out at a wavelength of 450 nm.

### 3.7. Chlorophyll Content

The chlorophyll content was measured with the method described by [58,59], with slight modifications. Briefly, about 50 mg of sample were extracted in two cycles using an 80% solution of acetone in water as a solvent. After 15 min of stirring, the extracts were centrifuged, and the supernatant was stored for their measurement. The spectrophotometric reading was performed in quartz cuvettes at two wavelengths: 663 and 647. The concentration of chlorophyll a and b was calculated using these equations:

Ca = 12,21 Abs663–2,81 Abs647

Cb = 20,13 Abs647–5,03 Abs663

and expressed as mg/100 g D.W.

### 3.8. Statistical Analysis

Measurements were performed in triplicates. A one-way analysis of variance (ANOVA) was used for evaluation. Tukey’s test with 95% confidence level was applied. Hierarchical Clustering Analysis (HCA) was used to process data from HSPME-GC/MS analysis. Statistical analysis was performed using Minitab ver. 19.0 and Microsoft Excel 365 software.

## 4. Conclusions

PAW treatment was shown to increase the product’s microbiological stability with a reduced energy consumption. This research, for the first time, studied the impact of PAW technology on the volatile components, phytosterols, carotenoids, and chlorophyll contents of rocket leaves. On one side, the observed changes were attributed mainly to oxidative reactions promoted by the reactive species present in PAW. However, an increase in carotenoids and chlorophyll b was reported after the shortest treatment time, indicating a higher extractability of these components, probably due to cellular structure breakdown. Hence, the content of the different compounds was the result of the balance between higher extraction and degradation due to possible oxidation. The effect of PAW on the selective chemical compounds’ results is therefore quite complex and not proportional to treatment time. Using intermediate processing time, PAW-10 could be recommended, since it allowed higher microbial decontamination while maintaining a volatile profile similar to that of the control; furthermore, the phytosterol content was improved while non-significant changes in carotenoids and chlorophyll contents were detected. Future studies are required to evaluate the effect of PAW on the other main bioactive compounds, in particular to better understand the effect of the processing parameters on the rocket’s nutritional quality.

## Figures and Tables

**Figure 1 molecules-26-07691-f001:**
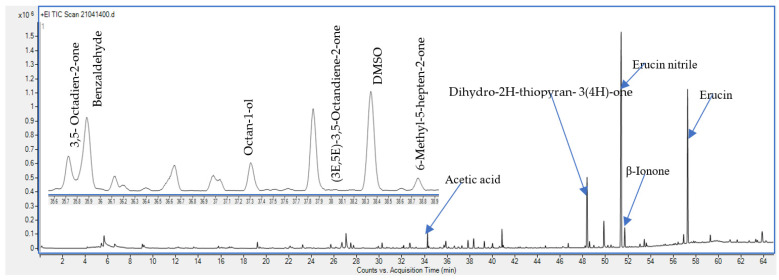
Representative chromatogram of the control rocket samples showing the major detected VOCs.

**Figure 2 molecules-26-07691-f002:**
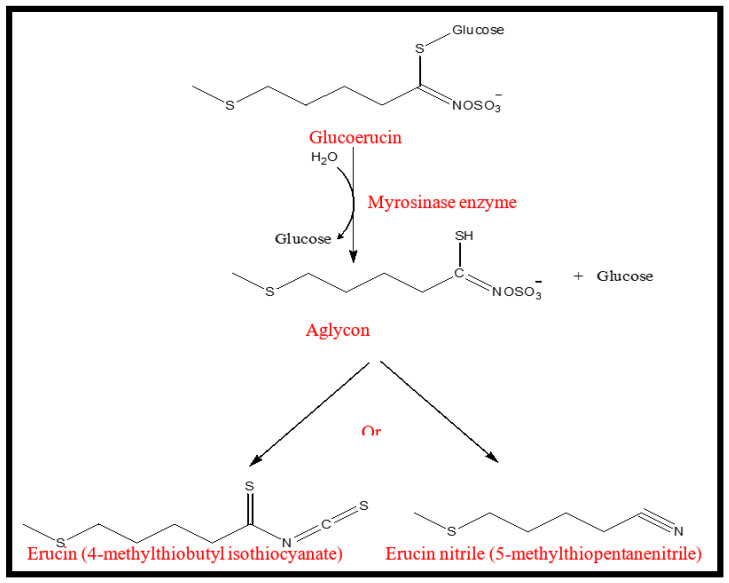
Enzymatic hydrolysis of glucoerucin by myrosinase enzyme intro erucin and erucin nitrile.

**Figure 3 molecules-26-07691-f003:**
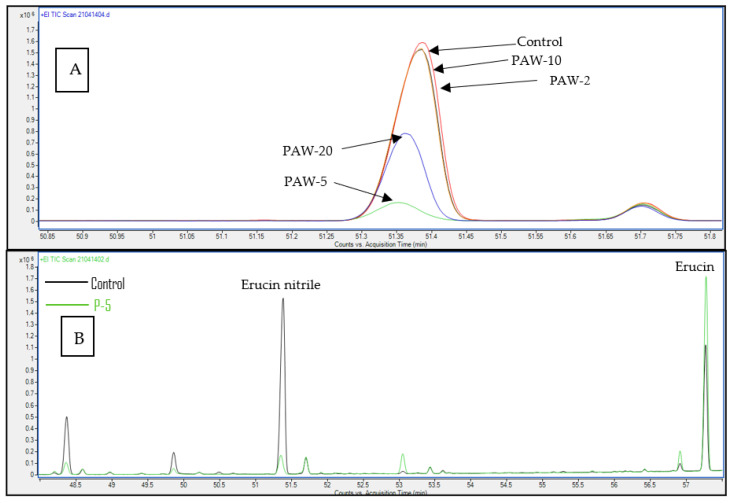
(**A**): Relative abundance of erucin nitrile in control and PAW-treated samples, (**B**): Relative abundance of erucin and erucin nitrile in control and PAW-5-treated sample.

**Figure 4 molecules-26-07691-f004:**
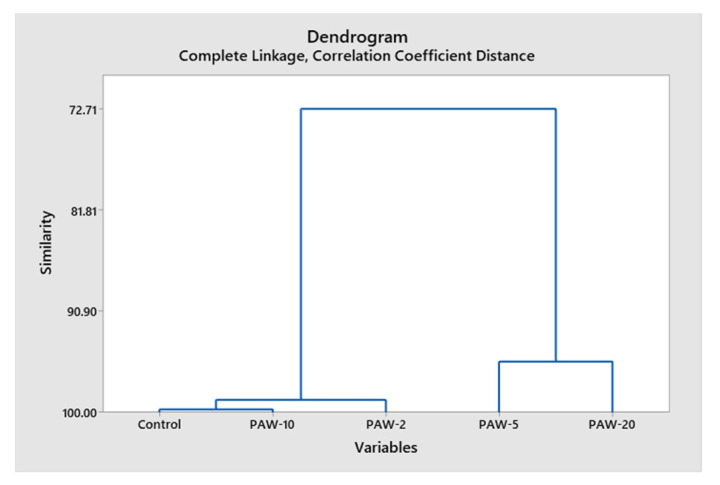
Dendrogram obtained from hierarchical clustering analysis (HCA) based on relative compositions of 52 VOCs detected by HS-SPME GC/MS in the control and PAW-treated rocket-salad samples at different processing times.

**Figure 5 molecules-26-07691-f005:**
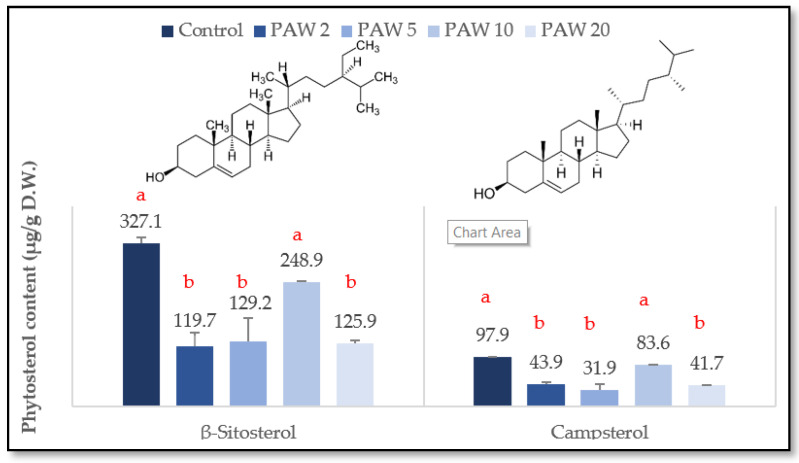
Changes in β-sitosterol and campesterol content of PAW treated rocket-salad samples at different processing times compared to control samples. Means that do not share letters for each compound differ significantly (*p* < 0.05) according to Tukey’s test. Legends: PAW-2, PAW-5, PAW-10, and PAW-20 refer to rocket samples subjected to plasma activated water (PAW) treatment for 2, 5, 10, and 20 min, respectively.

**Figure 6 molecules-26-07691-f006:**
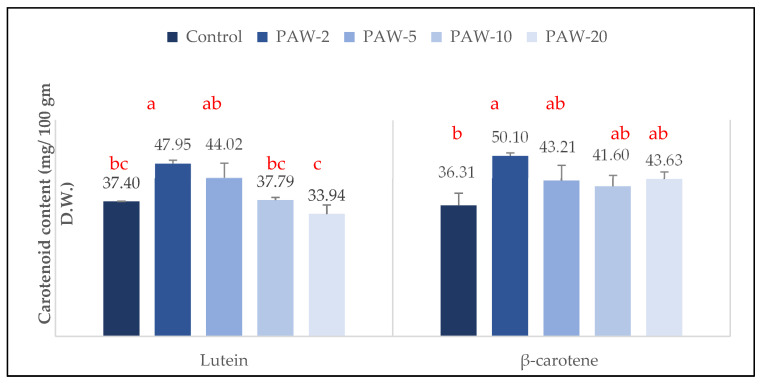
Changes in β-carotene and lutein contents of PAW treated rocket-salad samples at different processing times compared to control samples. Means that do not share letters for each compound differ significantly (*p* < 0.05) according to Tukey’s test. Legends: PAW-2, PAW-5, PAW-10, and PAW-20 refer to rocket samples subjected to plasma activated water (PAW) treatment for 2, 5, 10, and 20 min, respectively.

**Figure 7 molecules-26-07691-f007:**
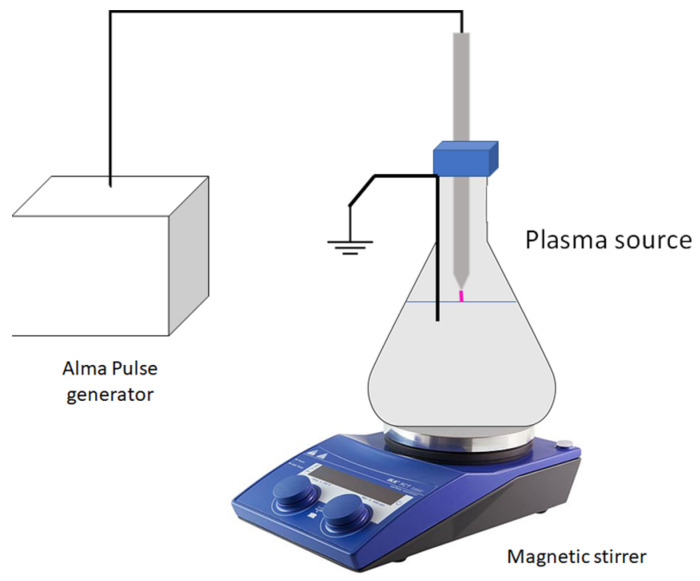
Schematic of the experimental setup of the corona source during the production of plasma activated water.

**Table 1 molecules-26-07691-t001:** Relative abundance of volatile organic compounds detected in rocket by HPSE-GC–MS before and after PAW treatment at different processing times.

	Compounds	Control	PAW-2	PAW-5	P-10	P-20	RI
	**Glucosinolate hydrolysis products (GHPs)**	**54.58 ± 1.48 ^a^**	**57.02 ± 0.08 ^a^**	**49.61 ± 3.85 ^a^**	**53.87 ± 4.44 ^a^**	**60.44 ± 3.15 ^a^**	
**1**	Methyl thiocyanate	0.04 ± 0.01 ^c^	0.09 ± 0.01 ^bc^	0.17 ± 0.04 ^a^	0.07 ± 0.01 ^bc^	0.12 ± 0.01 ^ab^	1282
**2**	5-Methyl Hexanenitrile	0.09 ± 0.01 ^a^	0.10 ± 0.01 ^a^	Nd ± 0.00 ^c^	0.11 ± 0.01 ^a^	0.03 ± 0.01 ^b^	1362
**3**	Heptanonitrile	0.18 ± 0.04 ^a^	0.18 ± 0.01 ^a^	Nd ± 0.00 ^c^	0.15 ± 0.01 ^a^	0.07 ± 0.01 ^b^	1403
**4**	1-Butene 4-isothiocyanate	0.07 ± 0.01 ^b^	0.14 ± 0.02 ^b^	0.56 ± 0.10 ^a^	0.13 ± 0.04 ^b^	0.42 ± 0.04 ^a^	1452
**5**	4-Methylthio butanenitrile	0.35 ± 0.11 ^ab^	0.38 ± 0.08 ^ab^	Nd ± 0.00 ^c^	0.45 ± 0.08 ^a^	0.11 ± 0.01 ^bc^	1784
**6**	Erucin nitrile	37.52 ± 2.54 ^a^	34.52 ± 1.60 ^a^	4.71 ± 0.27 ^c^	36.91 ± 5.41 ^a^	21.74 ± 0.06 ^b^	1935
**7**	Erucin	16.35 ± 4.16 ^b^	21.63 ± 0.74 ^b^	44.17 ± 4.19 ^a^	16.06 ± 0.82 ^b^	37.96 ± 3.27 ^a^	2143
	**Sulphur compounds**	**19.15 ± 0.81 ^a^**	**13.23 ± 0.05 ^bc^**	**10.08 ± 1.51 ^c^**	**15.02 ± 1.03 ^b^**	**10.57 ± 1.11 ^c^**	
**8**	Methyl disulphide	0.92 ± 0.11 ^b^	0.62 ± 0.05 ^c^	1.47 ± 0.01 ^a^	0.54 ± 0.01 ^c^	1.14 ± 0.05 ^b^	746
**9**	Dimethyl sulphide	3.30 ± 1.56 ^a^	3.32 ± 0.35 ^a^	3.22 ± 1.20 ^a^	3.57 ± 0.01 ^a^	3.82 ± 0.57 ^a^	760
**10**	Dimethyl trisulphide	0.18 ± 0.04 ^a^	0.08 ± 0.01 ^a^	0.33 ± 0.05 ^a^	0.14 ± 0.04 ^a^	0.07 ± 0.01 ^a^	1385
**11**	Dimethyl Sulfoxide	1.37 ± 0.19 ^a^	1.86 ± 0.18 ^a^	1.91 ± 0.21 ^a^	1.77 ± 0.13 ^a^	1.66 ± 0.11 ^a^	1577
**12**	Dihydro-2*H*-thiopyran-3(4*H*)-one	13.09 ± 2.64 ^a^	6.98 ± 0.56 ^bc^	2.79 ± 0.12 ^c^	8.76 ± 0.82 ^ab^	3.49 ± 0.37 ^c^	1845
**13**	Dimethyl sulfone	0.31 ± 0.01 ^a^	0.39 ± 0.04 ^a^	0.38 ± 0.04 ^a^	0.25 ± 0.05 ^a^	0.40 ± 0.03 ^a^	1899
	**Ketones**	**10.20 ± 0.81 ^b^**	**13.85 ± 0.73 ^ab^**	**15.48 ± 1.66 ^a^**	**11.64 ± 0.72 ^ab^**	**11.36 ± 0.58 ^b^**	
**14**	2,5-Dimethyl-3-hexanone	0.04 ± 0.01 ^bc^	Nd ± 0.00 ^d^	0.20 ± 0.01 ^a^	0.12 ± 0.02 ^b^	0.10 ± 0.04 ^b^	1188
**15**	3-Hydroxybutan-2-one	0.06 ± 0.01 ^a^	0.05 ± 0.00 ^a^	0.07 ± 0.01 ^a^	0.05 ± 0.01 ^a^	0.05 ± 0.01 ^a^	1297
**16**	1-Hydroxypropan-2-one	0.08 ± 0.02 ^ab^	0.05 ± 0.01 ^ab^	0.13 ± 0.04 ^a^	0.03 ± 0.00 ^b^	0.06 ± 0.02 ^ab^	1309
**17**	6-Methyl-5-hepten-2-one	0.41 ± 0.01 ^b^	0.95 ± 0.10 ^a^	0.46 ± 0.05 ^b^	0.52 ± 0.09 ^b^	0.39 ± 0.07 ^b^	1344
**18**	3-Octen-2-one	0.05 ± 0.01 ^b^	0.12 ± 0.01 ^b^	0.28 ± 0.04 ^a^	0.10 ± 0.02 ^b^	0.09 ± 0.00 ^b^	1409
**19**	3,5-Octadien-2-one	0.20 ± 0.07 ^b^	0.74 ± 0.01 ^ab^	0.82 ± 0.21 ^a^	0.64 ± 0.09 ^ab^	0.85 ± 0.23 ^a^	1514
**20**	(3*E*,5*E*)-3,5-Octandiene-2-one	0.79 ± 0.41 ^a^	1.27 ± 0.08 ^a^	1.30 ± 0.15 ^a^	1.00 ± 0.07 ^a^	1.28 ± 0.11 ^a^	1565
**21**	6-Methyl-3,5-heptadien-2-one	0.18 ± 0.04 ^a^	0.24 ± 0.00 ^a^	0.34 ± 0.08 ^a^	0.24 ± 0.08 ^a^	0.15 ± 0.01 ^a^	1587
**22**	(*E*)-β-Ionone	2.73 ± 0.30 ^b^	2.96 ± 0.21 ^b^	3.70 ± 0.02 ^a^	2.75 ± 0.07 ^b^	3.11 ± 0.04 ^ab^	1945
**23**	β-Ionone-5,6-epoxide Norisoprenoid	0.97 ± 0.04 ^c^	1.21 ± 0.06 ^ab^	1.27 ± 0.07 ^ab^	1.40 ± 0.04 ^a^	1.12 ± 0.04 ^bc^	1999
**24**	6,10,14-Trimethylpentadecan-2-one	1.61 ± 0.54 ^b^	1.75 ± 0.04 ^ab^	3.12 ± 0.59 ^a^	1.27 ± 0.07 ^b^	1.39 ± 0.01 ^b^	2129
**25**	(*E*)-geranylacetone	0.99 ± 0.25 ^ab^	1.64 ± 0.24 ^a^	1.09 ± 0.08 ^ab^	1.00 ± 0.09 ^ab^	0.85 ± 0.01 ^b^	1852
**26**	Dihydroactinidiolide Norisoprenoid	2.12 ± 0.13 ^ab^	2.87 ± 0.24 ^a^	2.73 ± 0.32 ^ab^	2.56 ± 0.06 ^ab^	1.95 ± 0.10 ^b^	2371
	**Aldehydes**	**6.48 ± 0.13 ^b^**	**6.84 ± 0.30 ^b^**	**12.71 ± 0.07 ^a^**	**7.75 ± 0.96 ^b^**	**8.00 ± 0.88 ^b^**	
**27**	2-Methyl propanal	0.87 ± 0.10 ^b^	1.14 ± 0.08 ^b^	2.26 ± 0.14 ^a^	1.20 ± 0.17 ^b^	1.22 ± 0.22 ^b^	810
**28**	2-Methyl butanal	0.77 ± 0.01 ^b^	0.56 ± 0.02 ^c^	1.53 ± 0.03 ^a^	0.64 ± 0.05 ^bc^	0.69 ± 0.06 ^bc^	911
**29**	3-Methyl butanal	0.77 ± 0.15 ^ab^	0.34 ± 0.04 ^c^	0.79 ± 0.02 ^a^	0.45 ± 0.10^bc^	0.47 ± 0.03 ^bc^	915
**30**	Pentanal	0.11 ± 0.01 ^b^	0.12 ± 0.01 ^b^	0.19 ± 0.02 ^a^	0.16 ± 0.01 ^ab^	0.11 ± 0.02 ^b^	976
**31**	Hexanal	0.15 ± 0.04 ^b^	0.19 ± 0.01 ^b^	0.33 ± 0.05 ^a^	0.18 ± 0.04 ^b^	0.10 ± 0.00 ^b^	1088
**32**	2-hexenal (*E*)	0.41 ± 0.02 ^a^	0.44 ± 0.06 ^a^	0.78 ± 0.32 ^a^	0.65 ± 0.21 ^a^	0.32 ± 0.13 ^a^	1236
**33**	Octanal	0.11 ± 0.01 ^bc^	0.15 ± 0.01 ^bc^	0.29 ± 0.02 ^a^	0.23 ± 0.05 ^ab^	0.23 ± 0.01 ^ab^	1299
**34**	Nonanal	0.83 ± 0.10 ^bc^	0.62 ± 0.03 ^c^	0.92 ± 0.02 ^ab^	0.61 ± 0.02 ^c^	1.12 ± 0.10 ^a^	1395
**35**	3-Furfural	0.17 ± 0.04 ^c^	0.18 ± 0.03 ^c^	0.85 ± 0.05 ^a^	0.16 ± 0.10 ^c^	0.55 ± 0.08 ^b^	1459
**36**	Benzaldehyde	1.15 ± 0.05 ^c^	1.62 ± 0.01 ^bc^	2.33 ± 0.01 ^a^	1.79 ± 0.05 ^ab^	1.28 ± 0.33 ^bc^	1518
**37**	β-cyclocitral	0.59 ± 0.01 ^b^	0.81 ± 0.05 ^ab^	1.00 ± 0.07 ^a^	0.74 ± 0.17 ^ab^	0.73 ± 0.01 ^ab^	1619
**38**	Benzene acetaldehyde	0.39 ± 0.02 ^a^	0.49 ± 0.19 ^a^	0.66 ± 2.16 ^a^	0.42 ± 0.11 ^a^	0.58 ± 0.06 ^a^	1635
**39**	2-Methyl benzaldehyde	0.20 ± 0.04 ^a^	0.22 ± 0.09 ^a^	0.83 ± 0.32 ^a^	0.55 ± 0.13 ^a^	0.63 ± 0.06 ^a^	1643
	**Fatty acids and esters**	**2.16 ± 0.07 ^b^**	**1.66 ± 0.13 ^b^**	**3.99 ± 0.34 ^a^**	**1.96 ± 0.30 ^b^**	**2.52 ± 0.11 ^b^**	
**40**	Acetic acid	1.23 ± 0.31 ^b^	1.09 ± 0.04 ^b^	2.37 ± 0.25 ^a^	1.23 ± 0.23 ^b^	1.71 ± 0.07 ^ab^	1447
**41**	Propanoic acid	0.05 ± 0.01 ^b^	0.06 ± 0.00 ^ab^	0.11 ± 0.02 ^a^	0.08 ± 0.02 ^ab^	0.08 ± 0.01 ^ab^	1530
**42**	Hexanoic acid	0.24 ± 0.04 ^b^	0.33 ± 0.04 ^ab^	0.54 ± 0.11 ^a^	0.23 ± 0.04 ^b^	0.44 ± 0.05 ^ab^	1840
**43**	Methyl palmitate	0.65 ± 0.21 ^ab^	0.19 ± 0.05 ^c^	0.99 ± 0.04 ^a^	0.43 ± 0.01 ^bc^	0.30 ± 0.01 ^bc^	2221
	**Alcohols**	**1.14 ± 0.10 ^b^**	**1.61 ± 0.08 ^a^**	**1.23 ± 0.08 ^ab^**	**1.57 ± 0.13 ^a^**	**1.39 ± 0.11 ^ab^**	
**44**	Pent-1-en-3-ol	0.12 ± 0.01 ^b^	0.31 ± 0.02 ^a^	0.14 ± 0.01 ^b^	0.25 ± 0.07 ^ab^	0.17 ± 0.01 ^b^	1180
**45**	pentan-1-ol	0.06 ± 0.01 ^b^	0.10 ± 0.01 ^a^	0.08 ± 0.01 ^ab^	0.06 ± 0.01 ^b^	0.07 ± 0.01 ^ab^	1267
**46**	(*Z*)-2-penten-1-ol	0.04 ± 0.01 ^c^	0.10 ± 0.01 ^b^	0.07 ± 0.01 ^bc^	0.16 ± 0.03 ^a^	0.06 ± 0.01 ^bc^	1329
**47**	Hexan-1-ol	0.05 ± 0.01 ^b^	0.09 ± 0.00 ^ab^	0.11 ± 0.01 ^a^	0.08 ± 0.01 ^ab^	0.10 ± 0.01 ^a^	1360
**48**	Hex-3-ene -1-ol	0.27 ± 0.10 ^b^	0.38 ± 0.01 ^ab^	0.26 ± 0.02 ^b^	0.51 ± 0.01 ^a^	0.20 ± 0.04 ^b^	1388
**49**	Octan-1-ol	0.40 ± 0.04 ^b^	0.34 ± 0.02 ^b^	0.50 ± 0.03 ^ab^	0.36 ± 0.04 ^b^	0.65 ± 0.07 ^a^	1552
**50**	Nonan-1-ol	0.08 ± 0.01 ^ab^	0.11 ± 0.01 ^a^	0.09 ± 0.01 ^a^	0.07 ± 0.01 ^ab^	0.04 ± 0.01 ^b^	1654
**51**	Phenylethyl alcohol	0.14 ± 0.02 ^b^	0.20 ± 0.01 ^a^	Nd ± 0.00 ^c^	0.09 ± 0.02 ^b^	0.10 ± 0.01 ^b^	1913
	**Alkanes**	**0.16 ± 0.01 ^c^**	**0.12 ± 0.06 ^c^**	**0.95 ± 0.00 ^a^**	**0.40 ± 0.03 ^b^**	**0.47 ± 0.05 ^b^**	
**52**	Undecane	0.16 ± 0.01 ^c^	0.12 ± 0.06 ^c^	0.95 ± 0.01 ^a^	0.40 ± 0.03 ^a^	0.47 ± 0.05 ^b^	1094

Means that do not share letters in the same row differ significantly (*p* < 0.05) according to Tukey’s test. Legends: PAW-2, PAW-5, PAW-10, and PAW-20 refer to rocket samples subjected to plasma activated water (PAW) treatment for 2, 5, 10, and 20 min, respectively.

**Table 2 molecules-26-07691-t002:** Changes in chlorophyll a (Chla), b (Chlb), and total chlorophyll contents of PAW treated rocket-salad samples at different processing times compared to control samples.

	Control	PAW-2	PAW-5	PAW-10	PAW-20
Chla	145.0 ± 4.1 ^a^	114.4 ± 15.2 ^ab^	111.4 ± 21.6 ^ab^	131.6 ± 9.3 ^ab^	102.4 ± 8.4 ^b^
Chlb	98.5 ± 13.4 ^b^	121.9 ± 17.3 ^a^	95.9 ± 11.8 ^b^	111.6 ± 3.3 ^ab^	100.4 ± 14.4 ^ab^
Total	243.4 ± 17.1 ^a^	236.3 ± 32.3 ^a^	207.3 ± 32.1 ^b^	243.2 ± 12.6 ^a^	202.7 ± 22.8 ^b^

Means that do not share letters for each compound differ significantly (*p* < 0.05) according to Tukey’s test. Legends: PAW-2, PAW-5, PAW-10, and PAW-20 refer to rocket samples subjected to plasma activated water (PAW) treatment for 2, 5, 10, and 20 min, respectively.

## Data Availability

Not applicable.

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
