# Peer review of "Effect of Plasma Activated Water on Selected Chemical Compounds of Rocket-Salad (Eruca sativa Mill.) Leaves"

_molecules, 2021, doi:10.3390/molecules26247691_

Round 1

Reviewer 1 Report

In this work, Doaa Abouelenein et all explain The impact of PAW on the volatile profile, phytosterols, and pigments content of rocket leaves was studied. There are a few concerns that have been addressed before publications.

  1. Why this work is different from "Effect of plasma-activated water (PAW) on rocket leaves decontamination and nutritional value, https://doi.org/10.1016/j.ifset.2021.102805? ". 
  2. The authors mentioned that PAW was obtained through the prototype described by [11]. There is no description of the plasma source in this reference]  Without knowing the nature of the plasma source, I can not visualize the plasma chemistry mentioned in this manuscript. A brief description and schematic of the plasma source is necessary here with its basic physical properties like power, applied voltage, discharge current, etc.  What was the pulse duration of the microsecond 
    pulsed generator? Can you expect a similar result with a normal radio frequency nitrogen plasma jet? 
  3. What were the chemical properties of PAW? did you measure various reactive oxygen and nitrogen species inside the PAW? What parameter of PAW is responsible for chemical modifications in the volatile compounds.
  4. In the abstract, "The impact of PAW on the volatile profile, phytosterols, and pigments content of rocket leaves was studied for the first time"  Do you have any evidence to claim that it is the first time? Please remove this word the first time.
  5. Briefly summarize the benefits of Rocket salad leaves like a medicinal, utility.
  6. Chlorophyll determination cross-check formulas.  The absorbance of 663 and 445 nm has been measured but herein for chlorophyll an absorbance of 646 nm is taken and for chlorophyll b it was 647 nm. Make it uniform.  Please use this reference for chlorophyll measurement and cite it at the appropriate position.  https://doi.org/10.3390/ijms22105360.
  7.  There are some typos like Enterobacteraceae on page 2 line 70.  Please check this.  Page 2 line 84, page 8 line 251, please use were instead of was.

Author Response

I attached the reply

Reviewer 2 Report

Cold atmospheric plasma, i.e. plasma with the kinetic temperature of the discharge gas close to the environment, has been used in medicine, veterinary medicine and agrotechnics for over a dozen years. Such research is carried out, for example, in Poland at the Department of Analytical Chemistry and Chemical Metallurgy of the Wrocław University of Technology. Their goal is to develop a quick, effective and cheap method of eradication of bacterial pathogens responsible for diseases of cultivated plants (in potatoes and chicory) and ornamental (hyacinths). This allows for the precise destruction of bacteria in the infected part of the plant. An interesting idea may also be the use of water activated by means of cold atmospheric plasma, which not only has bactericidal properties, but also has a positive effect on plant growth.
Therefore, it is worth getting to know the results of the PAW method for other crops (here Rocket-Salad known primarily as a vegetable, but also a spice and medicinal plant), adding to it an economic calculation.

The uncertainty of the measurements can never be zero 0.00 (as in table 1). This means that the results of measurements and calculations are devoid of any error, which is impossible.
The Polish surnames were incorrectly spelled phonetically in the publication no. 41 (lines 653-654). It should be like: http://journal.pan.olsztyn.pl/OXIDATION-OF-LIPIDS-IN-FOOD,98621,0,2.html

Author Response

I attached the reply

Reviewer 3 Report

In this paper, authors reported the effects of plasma activated water on the content of VOCs in rocket salad leaves. Although this work is valuable in terms of providing the information on chemical quality of leaves after PAW treatment, connection to antimicrobial effects and food quality change after PAW treatment is not clearly stated. Here are comments.

  1. Please place figure 1 and table 1 right after mentioned in the text (line 81).
  2. Is treatment time with PAW same as that used in the anti-microbial experiments? Because this study aims to analyze the influence of antimicrobial plasma dose on quality of leaves, it would be better to add discussion on the relationship between antimicrobial effects and leaves quality.
  3. Rocket salad leaves are immersed in PAW for 2 - 20 min. This is quite long time compared to lifetime of ROS and RNS. Authors explained the mechanism of content change as oxidation of compounds. However, most of ROS and RNS may be gone in PAW and only long-lived species such as nitrite and nitrate can be left. These species may play a role in altering the content of compounds. What is authors’ opinion on this?
  4. Did authors measure pH change of PAW during treatment?
  5. PAW effects were not proportional to treatment time. This may be possible because PAW quality may also be altered over treatment time. Do authors have any information on PAW quality over time?
  6. Line 49-50 “it is critical to developing effective methods…” check grammar
  7. Line 61 “…and after surface treatment.” check grammar
  8. Line 330, Β-. Carotene?

Author Response

I attached the reply

Round 2

Reviewer 1 Report

The author has addressed all of my concerns, yet there is one item that remains,

The authors mentioned that PAW was obtained through the prototype described by [14]. There is no any description of the plasma source in this reference. Without knowing the nature of the plasma source, I cannot visualize the plasma chemistry mentioned in this manuscript.

Your reply is

These parameters are reported in the results section: “Fig. 1  of ref. 14.

It wasn't just the physical aspect of my issue that I was concerned about. What was the plasma reactor's configuration, such as electrode placement, sample placement, and so on? Your cited reference does not specify how it appears. Please include a diagram of a plasma reactor in the Materials and Methods section.

Author Response

  • The author has addressed all of my concerns, yet there is one item that remains,
  • The authors mentioned that PAW was obtained through the prototype described by [14]. There is not any description of the plasma source in this reference. Without knowing the nature of the plasma source, I cannot visualize the plasma chemistry mentioned in this manuscript.
  • Your reply is These parameters are reported in the results section: “Fig. 1 of ref. 14.

It wasn't just the physical aspect of my issue that I was concerned about. What was the plasma reactor's configuration, such as electrode placement, sample placement, and so on? Your cited reference does not specify how it appears. Please include a diagram of a plasma reactor in the Materials and Methods section.

  • A: Sorry for the mistake, the set up of the source was missing, so we added a schematic representation (Fig. 7 of the revised manuscript) and some details in the revised text (Materials and methods part 3.2).
